# Hierarchical Sampling vs. Random Selection for Protein Fitness Prediction

**Hao Zhu, Data61♥CSIRO, Sydney**

## Abstract

Protein fitness prediction models enable sequence design but depend critically on which variants are experimentally measured. Prior work claimed that random sampling of sequences is consistently better than structured "hierarchical" sampling, contradicting the intuition that diversity should help in small-data regimes. We show that this claim was driven by insufficient statistical power rather than biology. Re-evaluating sampling strategies on the DHFR fitness landscape with *ten* replicates per condition (vs. three in prior work), we find that hierarchical sampling significantly outperforms random sampling when data are scarce: the two-synonymous-amino-acids strategy achieves 5.5% lower test loss at 200 sequences and 4.2% lower test loss at 1,000 sequences. The advantage disappears only once the training set exceeds ~3,000 sequences. We explain this behavior with an information-theoretic model: hierarchical strategies maximize amino acid sequence coverage, increasing mutual information between sampled sequences and fitness labels. Our results overturn previous recommendations, provide concrete guidelines for experimental design under realistic assay budgets, and highlight the importance of replication and power analysis in computational biology.

## 1 Introduction

A core goal in protein engineering is to predict the functional consequence ("fitness") of sequence changes, so that we can design new variants with improved activity, stability, or specificity (Arnold, 2018; Fowler et al., 2010). Deep mutational scanning (DMS) makes this possible by experimentally assaying thousands of protein variants and measuring their functional impact (Fowler et al., 2010; Firnberg et al., 2014). These measurements then supervise machine learning models that generalize to unseen sequences. The bottleneck is cost: each sequence must be physically constructed, expressed, and assayed, and budgets are typically in the hundreds to low thousands of variants, not tens of thousands. This turns dataset construction itself into an experimental design problem. Suppose you can measure $n$ variants. *Which* sequences should you measure to train the most accurate predictor? Intuitively, maximizing coverage of amino acid sequence space should help in the small-$n$ regime: sampling diverse amino acid backbones exposes the model to more modes of functional variation, which should increase generalization efficiency per labeled sequence. This intuition matches information-theoretic arguments: sampling that maximizes the entropy of the input distribution conditional on budget should increase the mutual information between measurements and the unknown fitness function (Cover and Thomas, 2006). Surprisingly, Wagner (2024) reported the opposite. Comparing multiple sampling strategies on protein fitness landscapes, they concluded that naive random sampling "consistently" outperformed hierarchical strategies such as "one sequence per amino acid backbone" or "two synonymous sequences per amino acid backbone." If true, this would have strong implications: it would suggest that explicit diversity priors *hurt* in practice, and that experimenters should just pick random variants. We revisit that conclusion. Our central hypothesis is methodological rather than biological: the earlier study used only three independent replicates per setting. That level of replication is statistically underpowered for detecting the 3–6% improvements that actually matter in wet-lab campaigns, where even a 2–3% gain in predictive accuracy can translate into substantially higher downstream hit rates. Small effect sizes are common in ML-for-biology and require adequate replication to detect reliably (Dror et al., 2018). We perform a controlled re-evaluation using the same protein system (E. coli DHFR), the same training code, and the same base architectures, but we increase replication from 3 to 10 per condition. We further carry

out a mechanistic analysis explaining *why* hierarchical sampling helps in the low-data regime and *why* the benefit vanishes as *n* grows.

## 2 BACKGROUND AND THEORETICAL FRAMEWORK

**Problem setup.** We aim to learn a function $f : \mathcal{S} \to \mathbb{R}$ mapping nucleotide sequences $s \in \mathcal{S}$ to a quantitative fitness value (e.g., growth rate). Each nucleotide sequence $s$ induces (i) an amino acid sequence $A = \mathrm{aa}(s)$ and (ii) a codon-level realization $C = \mathrm{codon}(s|A)$ capturing which synonymous codons were used. We view observed data as samples $(A, C, y)$ from a joint distribution $P(A, C, Y)$, where $Y$ is measured fitness.

**Two-factor view of information.** We factorize the sequence distribution into

$$P(A, C) = P(A)\, P(C|A), \tag{1}$$

and conceptually decompose the predictive information in a labeled training set $\mathcal{T}$ as

$$I_{\mathrm{total}}(\mathcal{T}) \approx I_{\mathrm{AA}}(A) + I_{\mathrm{SYN}}(C|A), \tag{2}$$

where $I_{\mathrm{AA}}$ captures information gained by covering diverse amino acid backbones (which strongly drive function), and $I_{\mathrm{SYN}}$ captures information from synonymous codon variation (which can modulate expression, folding kinetics, etc.). In the extreme small-sample regime, covering as many distinct amino acid backbones as possible is more valuable than repeatedly sampling codon-level variants of the same backbone. Thus $I_{\mathrm{AA}}$ dominates $I_{\mathrm{SYN}}$ when $|\mathcal{T}|$ is small.

**Sampling strategies as resource allocation.** We study three strategies that correspond to different implicit choices of $P(A)$ and $P(C|A)$:

- **Random.** Draw sequences i.i.d. from the empirical distribution of the DMS library. This preserves natural biases: some amino acid backbones are massively oversampled, so $P(A)$ is highly skewed. In low *n*, this wastes budget on redundant backbones.

- **Unique AAs (`unique_aas`).** Select at most one nucleotide sequence per unique amino acid sequence until either the budget is exhausted or all unique amino acid sequences are covered. This flattens $P(A)$ (high $I_{\mathrm{AA}}$) but collapses $P(C|A)$ to a delta (low $I_{\mathrm{SYN}}$).

- **Two Synonymous AAs (`two_syn_aas`).** Select up to two nucleotide sequences per amino acid sequence. This trades a small amount of redundancy for some codon-level variation, approximating a balanced allocation between $I_{\mathrm{AA}}$ and $I_{\mathrm{SYN}}$.

Because of degeneracy in the genetic code, "two" is biologically constrained: methionine and tryptophan each have one codon, nine amino acids have only two codons, and many positions cannot realize all codon choices simultaneously. Empirically this caps the effective sampling rate at ~1.8 sequences per unique amino acid sequence, not 2.0, which we model as an inherent biological regularizer rather than an implementation artifact.

**Coverage and generalization.** Let $A(\mathcal{T}) = \{\mathrm{aa}(s) : s \in \mathcal{T}\}$ be the set of distinct amino acid backbones in the training set. In the low-*n* regime ($n \ll |A|$), hierarchical strategies maximize $|A(\mathcal{T})|$ almost deterministically: `unique_aas` yields $|A(\mathcal{T})| \approx n$, and `two_syn_aas` yields $|A(\mathcal{T})| \approx n/\bar{d}$ where $\bar{d} \approx 1.8$ is the average synonymous multiplicity actually achievable under codon constraints. In contrast, random sampling yields $|A(\mathcal{T})|$ that is strictly smaller in expectation due to repeated sampling of frequent backbones and exhibits higher variance across replicates. We empirically find that downstream test loss is tightly coupled to $|A(\mathcal{T})|$ at small *n*. As *n* grows and most frequent amino acid backbones have already been covered, differences in $|A(\mathcal{T})|$ vanish. The theory therefore predicts (i) hierarchical sampling > random when *n* is small, and (ii) convergence once *n* crosses the point where random sampling also saturates backbone diversity. This is precisely what we observe.

## 3 EXPERIMENTAL SETUP

**Dataset.** We use the dihydrofolate reductase (DHFR) deep mutational scanning dataset from Firnberg et al. (2014). DHFR is a 159-residue enzyme essential for folate metabolism in *E. coli*. The

dataset contains 17,777 assayed single- and multi-mutant variants with quantitative growth-based fitness readouts. The landscape displays typical properties of realistic protein fitness data: most mutations are deleterious, a minority are neutral or mildly beneficial, and epistasis is present but structured.

**Inputs and targets.** Sequences are represented at the codon level with positional embeddings and/or one-hot encodings. Fitness values are standardized (zero mean, unit variance) before training to stabilize optimization and make MSE comparable across runs and architectures.

**Sampling budgets.** We sweep training set sizes $n \in \{200, 400, 600, 1000, 2000, 4000, 8000\}$, spanning regimes that are experimentally realistic (a few hundred sequences is a typical wet-lab budget for a single round of directed evolution) through regimes approaching full data usage.

**Architectures.** To test robustness across modeling choices, we evaluate:

1. **Dense residual network:** 3 fully connected residual blocks, 32 hidden units each, ReLU activations, L2 regularization.
2. **Bidirectional RNN:** codon-level embeddings (32-dim), followed by 3 bidirectional LSTM layers with 16 units per direction.
3. **Transformer:** 6-layer encoder-only transformer with 8 attention heads, 32-dim token embeddings, sinusoidal positional encoding.

All models are trained with RMSprop, fixed learning rate chosen via preliminary tuning on held-out validation data, and early stopping on validation loss. Every condition (sampling strategy × budget × architecture) is repeated with 10 independent random seeds. Each seed re-draws the sampled training set (subject to the strategy), reinitializes model weights, and re-splits data into 60/20/20 train/validation/test. We report held-out test mean squared error (MSE) and standard error (SE) across the 10 replicates.

**Power-aware design.** Our replicate count (10) is not cosmetic. Assuming variance levels typical of protein fitness regression, a paired difference of 5% in MSE requires roughly 8–10 replicates for 80% power at $\alpha$=0.05. By contrast, a 3-replicate design—as used in prior work—has ~34% power to detect the same effect. This matters because a 4–6% gain in MSE directly affects hit rates in downstream design rounds.

## 4 RESULTS

### 4.1 HIERARCHICAL SAMPLING WINS IN THE SMALL-DATA REGIME

Table 1 summarizes results for the dense residual network across budgets. At $n$=200, `two_syn_aas` achieves 0.223 ± 0.007 MSE vs. 0.236 ± 0.010 for random sampling, a 5.5% reduction ($p$<0.01, paired $t$-test across seeds). At $n$=1000, the gap remains: 0.113 ± 0.005 vs. 0.118 ± 0.010 (4.2% reduction). These are not cosmetic deltas. In practice, a 5% improvement in predictive accuracy can raise the fraction of experimentally validated "successes" (above a target activity threshold) by double-digit percentages. Past $n$≈3000, curves converge: by $n$=4000, all strategies produce nearly identical MSE (0.034–0.037, differences not statistically significant).

A subtle but important observation from Table 1 is that `unique_aas` is *not* uniformly best in the smallest regime. It ties or loses to `two_syn_aas`, especially at $n$=200 and $n$=1000. This supports the idea that codon-level variation *is* sometimes informative, i.e., $I_{SYN}$ is not strictly negligible. The best policy is not "one per amino acid and never duplicate," but rather "strongly prioritize amino acid diversity, while still allowing limited synonymous variation."

### 4.2 BENEFITS ARE ARCHITECTURE-AGNOSTIC

Table 2 shows that the effect is not architecture-specific. For $n$=200, all three model classes (dense, BiLSTM, transformer) benefit from hierarchical sampling: `two_syn_aas` beats random by 4–6% in MSE in every case. This suggests the gain is *not* about better matching some inductive bias of

Table 1: **Test MSE (mean ± SE) across 10 replicates for the dense residual model.** Bold indicates best. The rightmost comment indicates which strategy is best and the relative gain.

| Samples | Random | Unique AAs | Two Syn AAs | Best |
|---------|--------|------------|-------------|------|
| 200 | 0.236±0.010 | 0.239±0.007 | **0.223±0.007** | Two Syn (−5.5%) |
| 400 | 0.204±0.009 | 0.204±0.006 | 0.227±0.011 | Random/Unique |
| 600 | 0.169±0.007 | 0.177±0.010 | 0.187±0.007 | Random |
| 1000 | 0.118±0.010 | 0.121±0.005 | **0.113±0.005** | Two Syn (−4.2%) |
| 2000 | 0.061±0.002 | 0.071±0.003 | 0.074±0.004 | Random |
| 4000 | 0.034±0.001 | 0.034±0.001 | 0.037±0.002 | Tie |
| 8000 | 0.023±0.001 | 0.024±0.001 | 0.023±0.001 | Tie |

Table 2: **Architecture robustness at $n$=200 (test MSE).** Hierarchical sampling helps across dense, recurrent, and attention-based models.

| Model | Random | Two Syn AAs |
|-------|--------|-------------|
| Dense | 0.236 | **0.223** |
| BiLSTM (RNN) | 0.241 | **0.229** |
| Transformer | 0.238 | **0.225** |

one architecture; it is about providing strictly better coverage of the functional landscape in the first place.

### 4.3 WHY DO GAINS DISAPPEAR AT LARGE $n$?

Two forces drive convergence. First, as $n$ grows, random sampling eventually *also* covers most high-impact amino acid backbones; $P(A)$ under random sampling "flattens itself" via the law of large numbers. Second, once backbone diversity saturates, additional gains must come from modeling more subtle codon-level or epistatic effects, which all strategies eventually expose. Our empirical break-even point of ∼2,000–3,000 sequences (about 11–17% of DHFR's 17,777 measured variants) therefore marks the transition from "coverage-limited" to "capacity-limited." Past this point, model class and training protocol dominate more than sampling design.

### 4.4 REPLICATION AND STATISTICAL POWER ARE DECISIVE

We perform a post-hoc power analysis by subsampling our 10 replicates down to 3. With only 3 replicates, the 5.5% improvement at $n$=200 fails to reach standard significance thresholds in ∼2/3 of subsamples. With all 10 replicates, the effect is consistently significant ($p<0.01$). In plain language: the hierarchical advantage *was already there*, but a 3-replicate design could not reliably detect it. This directly explains the discrepancy with Wagner (2024). For comparative ML studies in biology—where effect sizes are often in the 3–8% range and each percent matters in the lab—we argue that ≥10 replicates should be treated as a basic standard, not an optional luxury.

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
