# OpenReview forum: "Hierarchical Sampling vs. Random Selection for Protein Fitness Prediction"
_AJCAI/2025/Workshop/AIML-CEB — AIML-CEB 2025 Oral_

### Official Review · Reviewer_dwJo · 2025-11-07
**The importance of proper cross-validation**

**Rating:** 9
**Confidence:** 5

**Review:**

This is great work. I've heard a few criticisms of [Wagner, 2024], but this is the first formal dissection of the work I have seen, and I think it does a great job. This will make for a marvellous addition to the workshop, and I think it will generate a lot of discussion.

I don't have any criticisms about the methodology employed, but I do have some suggestions for future work or ways to expand on the analysis here.

CNNs are another common architecture I have seen in the literature for sequence fitness prediction tasks -- e.g. two layers with pooling to capture local and global structural effects. These are little easier and faster to train than RNNs (don't suffer from exploding gradients). It's not necessary, but it may be nice to include and example of these, e.g.:

- [Improving protein optimization with smoothed fitness landscapes](https://openreview.net/forum?id=rxlF2Zv8x0)
- [Variational Search Distributions](https://openreview.net/forum?id=1vrpdV9U3i)

There have been some theoretical analyses of predictors on "fitness landscapes" that attempt to quantify features such as "ruggedness" and "epistasis", and then formulate specific predictive architectures for one-shot mutational design, with generalisation guarantees:

- [Protein Fitness Landscape: Spectral Graph Theory Perspective](https://raw.githubusercontent.com/mlresearch/v258/main/assets/zhu25c/zhu25c.pdf)

Finally, depending on the final intentions and objectives, I wonder if held-out MSE is the best objective to analyse? Here you have studied which sampling strategies give rise to the best predictors. But I wonder if typically we are concerned about which predictors will yield the best selection of sequences to build and test (next)? Of course the two are linked, and if we have good initial libraries to train predictors, we would expect they will be more effective for subsequent selection tasks. However, selection is a binary choice, and so I wonder if it would be worth extending this analysis to that setting? For example, we may only be interested in sequences that are fitter than same incumbent, like the "wild-type", and so then measures like precision, recall, AUC, Spearman correlation (between predictions and measured fitness, since ranking would be more important than MSE), pass@K/precision@K (for batch selection) etc would be nice to quantify for some selection procedure (e.g. thresholding the predictions). If we are interested in finding "the fittest sequence", maybe we do not need a symmetric measure like MSE (since we only care about predicting/capturing high fitness sequences).

Following this line of thought, we can also take a decision theoretic view of this problem. A common and relevant paradigm for this would be the Bayesian optimisation/Bandits literature -- especially if we are able to do sequential experimentation, we can view this as a "black-box optimisation" problem. If you are interested in this, this is a worthwhile read:

- [Bayesian Optimization book](https://bayesoptbook.com/book/bayesoptbook.pdf)

And there is a host of work generalising this to sequence optimisation, see variational search distributions work above but also (to point out just a few):

- [Protein Design with Guided Discrete Diffusion](https://proceedings.neurips.cc/paper_files/paper/2023/file/29591f355702c3f4436991335784b503-Paper-Conference.pdf)
- [Amortized Active Generation of Pareto Sets](https://openreview.net/forum?id=jNQ40aw5qL&referrer=%5BAuthor%20Console%5D(%2Fgroup%3Fid%3DNeurIPS.cc%2F2025%2FConference%2FAuthors%23your-submissions))

I hope this feedback is useful, and thanks again for submitting this great work to the workshop!

---

### Official Review · Reviewer_bzLy · 2025-11-10
**Overturning findings comparing hierarchical sampling and random selection for exploring protein fitness landscapes**

**Rating:** 9
**Confidence:** 4

**Review:**

For small training datasets, prior work claimed that randomly sampling sequences outperforms hierarchical sampling for exploring the protein fitness landscape. The authors overturn this claim using a larger number of replicates, and describe an information-theoretic model to support their finding.

The paper is beautifully written. The introduction is compelling and clearly establishes the biological, experimental and statistical motivations for their study. Both the theoretical framework and experimental approach are systematically designed and described. The study underlines the importance of using sufficient replicates for statistical power, a seemingly simple yet apparently overlooked aspect of experimental design.

My main query is around the selection of the three NN architectures used to evaluate sampling strategies. What is the rationale for these specific architectures, are these the most commonly used for this specific problem, or should others also be considered? Are the number of model parameters comparable between the three, and does this effect the observed results? Similarly, what is the justification for using MSE for evaluation? Is this selected to match what was reported in (Wagner, 2024)?

---

### Official Review · Reviewer_XnGW · 2025-11-11
**Overturning a published result, the way science should work**

**Rating:** 8
**Confidence:** 4

**Review:**

## Summary
The manuscript revisits the question of how to sample a protein fitness landscape,
to understand a previous result claiming that random sampling is better than hierarchical sampling.
By increasing the number of replicates from three to ten, the authors show that hierarchical sampling
significantly outperforms random sampling for sparse data, overturning previous results.
The authors provide a discussion based on information gain why for small number of sequences,
hierarchical sampling performs better, and the effect reduces for larger number of sequences.

## Evaluation
I really like the premise of the paper, to carefully revisit previously published results.
The best part is that the authors show by careful arguments and clear empirical experiments
(over model, budgets, and mutations), that hierarchical sampling indeed does better in a certain regime.

Suggestions for improvement:
- It would be very useful to provide a plot of power vs replicate count, for a few different paired differences.
This would be informative for a reader to estimate how many replicates they need, given their belief
of the different differences between sampling methods.
- I am curious whether the analysis could be extended to more mutations, and perhaps the authors
may be able to analyse the combinatorially complete three amino acid setting in
https://doi.org/10.1126/science.adh3860

---

### Decision · Program_Chairs · 2025-11-12

Accept (Oral)